# Effective Training Methods for Cucumber Production in Newly Developed Nutrient Film Technique Hydroponic System

Nethone Samba [1,*] , Osamu Nunomura [2], Akimasa Nakano [1] and Satoru Tsukagoshi [1]

1 Center for Environment, Health and Field Sciences, Chiba University, Kashiwa-no-ha 6-2-1, Kashiwa, Chiba 277-0882, Japan
2 Japan Plant Factory Association, Kashiwa-no-ha 6-2-1, Kashiwa, Chiba 277-0882, Japan
* Correspondence: anatolesame@yahoo.fr

**Abstract:** Cucumber (*Cucumis sativus* L., cv. "Nina Z") plants grafted onto squash (*Cucurbita maxima*, cv. "Yu Yu Ikki") were grown in a greenhouse using the newly developed nutrient film technique (NFT) hydroponic system "Kappa land" (Mitsubishi aqua solutions Co., Ltd., Tokyo, Japan), from February to June 2022. The growth and development of cucumbers were examined under two different training methods: Lowering training (LT) and Pinching training (PT). Data collected were related to water and nutrient consumption, plant growth and development parameters, and the workload of the main activities. The results showed that plants grown under the LT recorded significantly higher total stem length (10.9 m) and number of nodes (133). In addition, from 21 April to 19 May, the leaf area index was significantly higher in the LT treatment. The highest total yield (15.4 kg m$^{-2}$) and marketable yield per unit area (13.8 kg m$^{-2}$) were recorded in the LT treatment. Regarding fruit growth, the fruits took 14 and 19 days to reach the standard harvest weight in the PT and LT treatments, respectively. In addition, the fruits were more straight in the PT treatment. The water use efficiency was not significantly different between the two treatments. However, the nutrient use efficiency was significantly higher in the PT treatment because plants produced more vegetative organs in the LT treatment instead of fruits. The work for removing old leaves and harvesting fruits was simplified in the LT treatment. The LT method can be effective for the automation of old leaf removal and fruit picking by the robot in the future.

**Keywords:** greenhouse; leaf area index; lowering training; pinching training; workload; yield

## 1. Introduction

Cucumber (*Cucumis sativus* L.) is an essential and commercially popular cucurbitaceous vegetable crop holding a coveted position in the vegetable market [1,2]. It is a rich source of valuable nutrients and bioactive compounds used not only as food but also in therapeutic medicine and cosmetology [3–5]. Cucumber is cultivated throughout the world under tropical and subtropical climates [6]. China, India, Turkey, Iran, Japan, and the United States are among the countries of commercial cucumber producers [7].

Cucumber productivity difference among countries is remarkable. In 2020, the world cucumber production was estimated to be 91.258 million tons on 2,261,318 hectares, achieving an average yield per square meter of 4.04 kilos [8]. The Netherlands had the highest yield, with 70.52 kg m$^{-2}$. The second best-performing country was Spain, with a yield of 10.28 kg m$^{-2}$, which is 585.99% lower than the Netherlands yield. Additionally, the recorded yields in 2020 were 8.01, 6.71, 5.04, and 2.25 kg m$^{-2}$, respectively, in Greece, Poland, Turkey, and Egypt [9]. In Japan, cucumber yield was estimated at 3.4 kg m$^{-2}$ and 10.7 kg m$^{-2}$ for summer-autumn and winter-spring cultivations, respectively [10].

The yield variability among countries can be explained by many factors, such as the cultivars, the production technologies, and the cultivation techniques used in each country. In advanced facility-growing countries such as the Netherlands, high-productivity

technologies such as soilless nutrient cultivation, environmental control technologies, and high-yield varieties are promoted [11]. However, those technologies and high-yield varieties are limited or unspread in other countries. Moreover, the cultivation methods of cucumber in middle and low-advanced facility-growing countries could be identified as the limited factors to yield improvement. According to Maeda and Ahn [11], the low yield of cucumber in Japan compared to the Netherlands is mainly related to the cultivation method. The authors analyzed Japanese greenhouse cucumber research and indicated that the most suitable method for cucumber cultivation has not yet been determined.

The pinching and the lowering cultivation methods are familiar in Japan [11]. With the pinching method, the side shoots are pinched several times to increase the number of lateral branches. The yield in this method varies depending on how the growers pluck the side shoots and the variety used. On the other hand, with the lowering cultivation method, some lateral branches (1–4) are maintained and trained to a wire placed at 1.5–2 m, and once the tips of the laterals are within 0–30 cm of the overhead wire, they are continuously lowered. This method is easy to understand and is often used for large-scale cultivation. However, the leaf area index could become higher, especially in hydroponics culture, and reduce plant productivity, depending on the number of lateral branches maintained. Ota et al. [12] reported that the high-wire cultivation method is used in the Netherlands to cultivate cucumber. The main stem is grown continuously, and the top of the stem is maintained at 0–40 cm below the wire, which hangs at 3.5–4 m. When the top of the plant is within 40 cm of the horizontal wire, the main stem is lowered by moving the stem sideways.

Mardhiana et al. [13] indicated that appropriate cultivation techniques are required to increase cucumber production. They reported that cultivation techniques could increase cucumber production through proper pruning.

Manipulation of canopy architecture through pruning and training with appropriate spatial arrangements has been identified as a key management practice for getting maximum, marketable yields from greenhouse crops [14]. Training improves a plant's ability to obtain the sunlight needed for growth [14,15], and adequate air movement around the plant reduces the risk of fungus and insect problems [14]. Moreover, pruning influenced vegetative growth and fruit quality [16,17] and had no adverse effect on peach [17] and cucumber [13].

The current study aims to propose the direction of future cucumber production technology by considering attractive cultivation methods for maximizing productivity. Specifically, two cultivation methods (lowering and pinching trainings) were compared. Furthermore, the characteristics of each training method were examined, and guidance was proposed to ease the spread of cucumber cultivation in hydroponics for small and medium enterprises.

## 2. Materials and Methods

### 2.1. Experimental Greenhouse

The experimental site is located at the Kashiwa-no-ha campus of Chiba University (35°53′32″ N latitude and 139°57′0″ E longitude at an altitude of 20 m above sea level). The experiment was carried out in a Venlo-type greenhouse with double spans, oriented East-West. The greenhouse (20 m length $\times$ 8 m width $\times$ 5 m height) is covered with a polyethylene film and has heaters and fans to adjust the air temperature. The heating system was set to turn on once the indoor air temperature fell to 14 °C. The daytime relative humidity of the greenhouse was maintained at 70% by using a fogging system (Ikeuchi Co. Ltd., Osaka, Japan). The $CO_2$ was supplied every morning to keep the indoor $CO_2$ concentration at around 400 $\mu$mol mol$^{-1}$. Air temperature, relative humidity, $CO_2$ concentration, and solar radiation in the greenhouse were recorded every 2 min by an agricultural production support system named Midori cloud (SERAKU Co., Ltd., Tokyo, Japan). In addition, the air temperature, $CO_2$ concentration, and relative humidity near the plant's canopy, at 1.5 m above the ground in each treatment, were recorded every 30 min using data loggers (TR-76Ui, T&D Corporation, Matsumoto, Japan).

### 2.2. Seedlings Production

The evaluated cucumber cultivar, "Nina Z," is characterized by its stable fruit shape and resistance to powdery mildew and brown spot. In addition, it has a good branching ability and a female flowering rate of almost 100% on the main vine. Each node of the main vine can bear 1–2 fruits. The plant vigor is maintained stable from the beginning to the end of the crop cycle.

In 2022, on 12 and 14 January, respectively, seeds of squash (*Cucurbita maxima*, cv. "Yu Yu Ikki," Saitama Progenitor Breeding Society Co., Ltd., Saitama, Japan) and cucumber (*Cucumis sativus* L., cv. "Nina Z", Saitama Progenitor Breeding Society Co., Ltd., Saitama, Japan) were sown into 128 cells cell-trays. The cell-trays were filled with a commercial growing substrate (Na-terra, Mitsubishi Chemical Agri Dream Co., Ltd., Tokyo, Japan). After germination in darkness at 28 °C, seedlings were raised in a nursery room equipped with fluorescent lamps (Nae Terrace, Mitsubishi Chemical Agri Dream Co., Ltd., Tokyo, Japan) for five days. The nursery room environmental conditions were set at 24/18 °C (light/dark period), with a photoperiod of 14 h and a $CO_2$ concentration of 1000 μmol mol$^{-1}$. A flood and drain hydroponic technique was used to supply the nutrient solution (EC 1.4 dS m$^{-1}$) to the seedlings once a day. On January 21, cucumber scions were grafted onto squash rootstocks using the hole insertion approach grafting.

First, the growing points of the rootstocks were wholly and carefully removed, and the hypocotyls were cut and drilled at about 30° angle using a bamboo gimlet. Next, the scion hypocotyls were cut at about 30° angle to form a 7–8 mm long wedge. The scions were then inserted into the prepared rootstock holes.

Grafted seedlings were transplanted into a cell-tray filled with commercial growing substrate and grown in a healing chamber (Nae Terrace, Mitsubishi Chemical Agri Dream Co., Ltd., Tokyo, Japan) for two weeks. The healing chamber environmental conditions were set at a temperature of 28 °C, relative humidity of 90%, and a photoperiod of 12 h. During the first week, grafted seedlings were placed in a healing box where the air was humidified for 4–5 min per hour to maintain the relative humidity above 90%. Above the healing box, fluorescent lamps were used to provide light at different intervals of time. Grafted seedlings were in the dark for the first 24 h, and the light was then gradually increased up to 300 μmol m$^{-2}$ s$^{-1}$ for 12 h per day. After the first week, the healed grafted seedlings were removed from the healing box and acclimated for one more week. The seedlings were bottom-irrigated once daily (10 min) with a nutrient solution (EC 1.4 dS m$^{-1}$).

### 2.3. Hydroponic System and Plants Density

In this study, the nutrient film technique (NFT) hydroponic system "Kappa land" (Mitsubishi aqua solutions Co., Ltd., Tokyo, Japan) was used to grow cucumbers. The NFT hydroponic system "Kappa land" uses less water, reducing the energy cost for heating or cooling the nutrient solution in winter and summer. Furthermore, the work for replanting is simplified.

Two cultivation beds were constructed with foam materials. Each cultivation bed had a dimension of 7.17 m in length, 0.45 m in width, and 12 cm in depth. The interior surfaces of the growth trays were covered with a net sheet and a plastic film to allow the nutrient solution to flow through the roots' zone. Under each cultivation bed, a reservoir (tank) with a capacity of 60–75 L was placed. The nutrient solution was pumped from the pool and injected into the growth trays. One part of the injected nutrient solution passes through a sprinkler tube and spry continuously on the plant roots. A float switch automatically adjusted the nutrient solution level in the pool. The fertilizers were pumped into the pool accordingly to the nutrient solution's electrical conductivity (EC). Plants were arranged at a density of 1.66 plants m$^{-2}$ with 17 plants per cultivation bed. Figure 1 shows the schematic diagram of the NFT hydroponic system used in the current study.

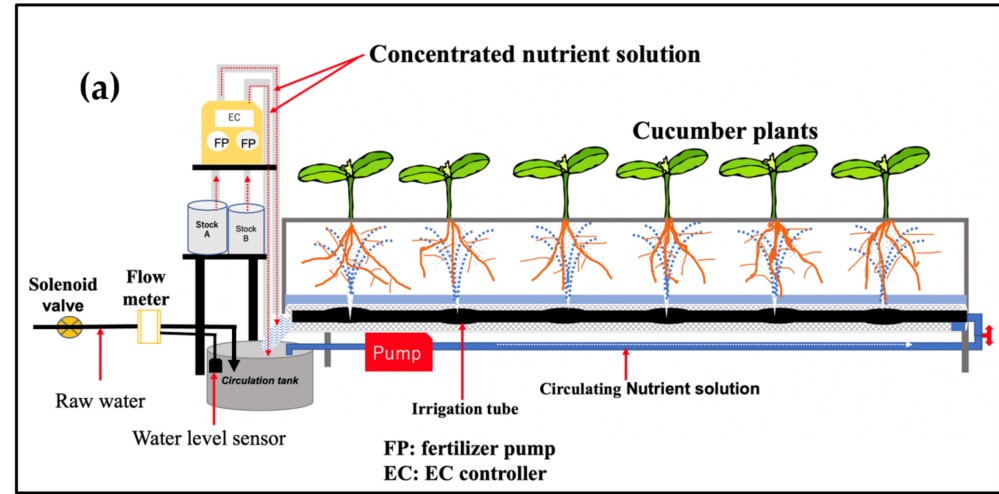

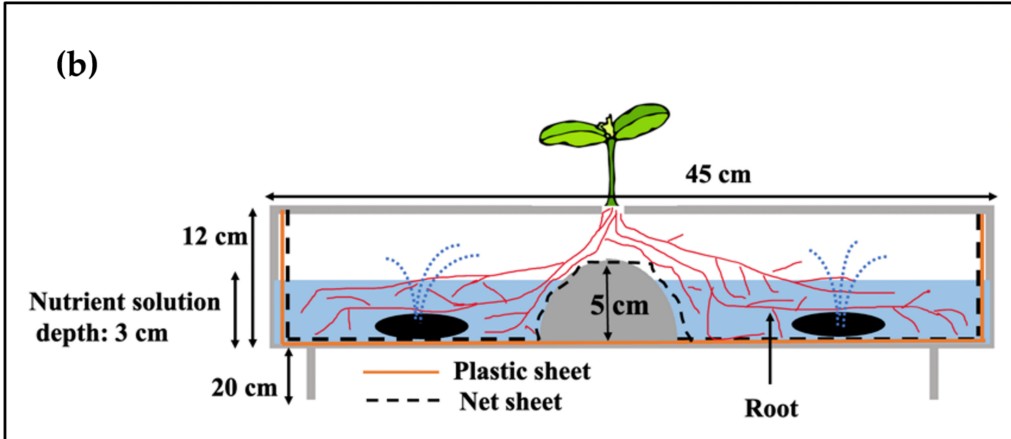

**Figure 1.** Schematic diagram of the NFT hydroponic system: lateral (**a**) and front (**b**) view.

### 2.4. Nutrient Solution

Two concentrated nutrient solutions, A and B, were formulated and stored in different tanks. The two nutrient solutions were diluted 100 times before usage. The chemical composition of the two concentrated nutrient solutions is shown in Table 1.

**Table 1.** Concentrated nutrient solution composition.

| Concentrated Solution A | |
| --- | --- |
| **Component** | **Concentration (g L$^{-1}$)** |
| Potassium nitrate ($KNO_3$) | 0.87 |
| Ammonium dihydrogen phosphate ($NH_4H_2PO_4$) | 0.02 |
| Magnesium sulfate heptahydrate ($MgSO_4 \cdot 7H_2O$) | 0.55 |
| Potassium sulfate ($K_2SO_4$) | 0.09 |
| Pre-mixed Micronutrients | 0.03 |
| Phosphoric acid ($H_3PO_4$) | 0.04 |
| Concentrated solution B | |
| Calcium nitrate tetrahydrate $Ca(NO_3)2 \cdot 4H_2O$ | 1.06 |

### 2.5. Experimental Treatments and Plant Care

The training and pruning methods applied to each plant are described as follows.

Pinching training (PT) method (Figure 2): the main stem was allowed to grow vertically following a string and was pinched at the stage of the 18th node. All flowers and laterals of the 1–5th nodes were removed. The other lateral branches were allowed to grow and were repeatedly pinched at the 1st or 2nd node depending on the lateral branch vigor and its position on the main stem.

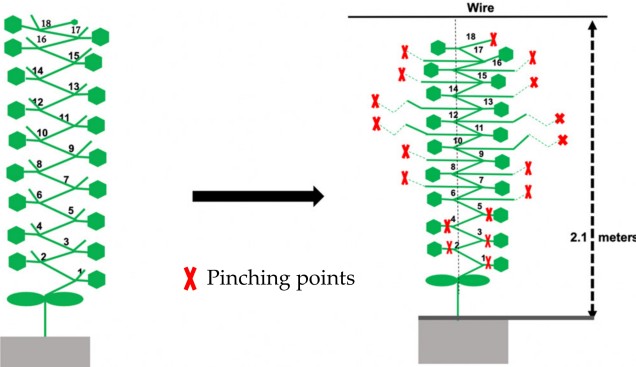

**Figure 2.** Pinching training method.

The lateral branches of the 6–9th nodes and the 14–18th nodes were pinched at the 1st node, while the lateral branches of the 10–13th nodes were pinched at the 2nd node.

Lowering training (LT) method (Figure 3): the main stem was allowed to grow vertically following a string and was pinched at the stage of the 16th node. All flowers and laterals of the 1–5th nodes were removed.

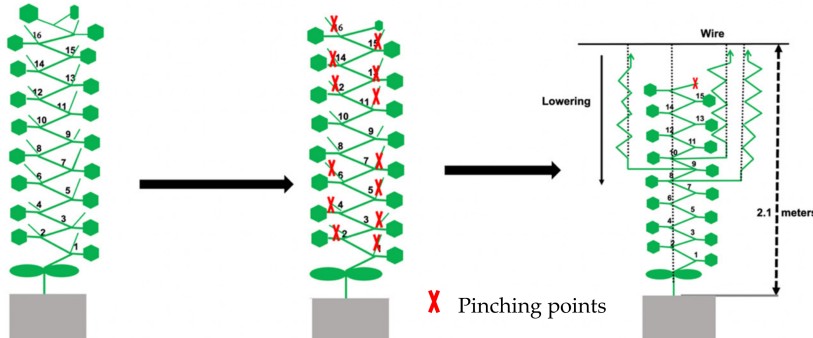

**Figure 3.** Lowering training method.

Three (3) selected lateral branches were allowed to grow vertically following strings and were trained onto two (2) overhead wires placed at 2.1 m above the cultivation bench. For each plant, two lateral branches were trained onto the left or right-side overhead wire and the third lateral branch on the other overhead wire. Once the three (3) selected lateral branches reached the overhead wire, they were continuously lowered during the rest of the growing period. The other lateral shoots were regularly pruned. The distance between lateral branches was maintained at 30 cm.

During the growing period, over-matured leaves and tendrils were frequently pruned. Different pesticides were used to prevent biotic diseases. In addition, the nutrient solution EC and temperature were maintained at an average of 2.4 dS m$^{-1}$ and 24.7 °C, respectively (Figure 4). The pH was maintained between 5.5 and 6.5.

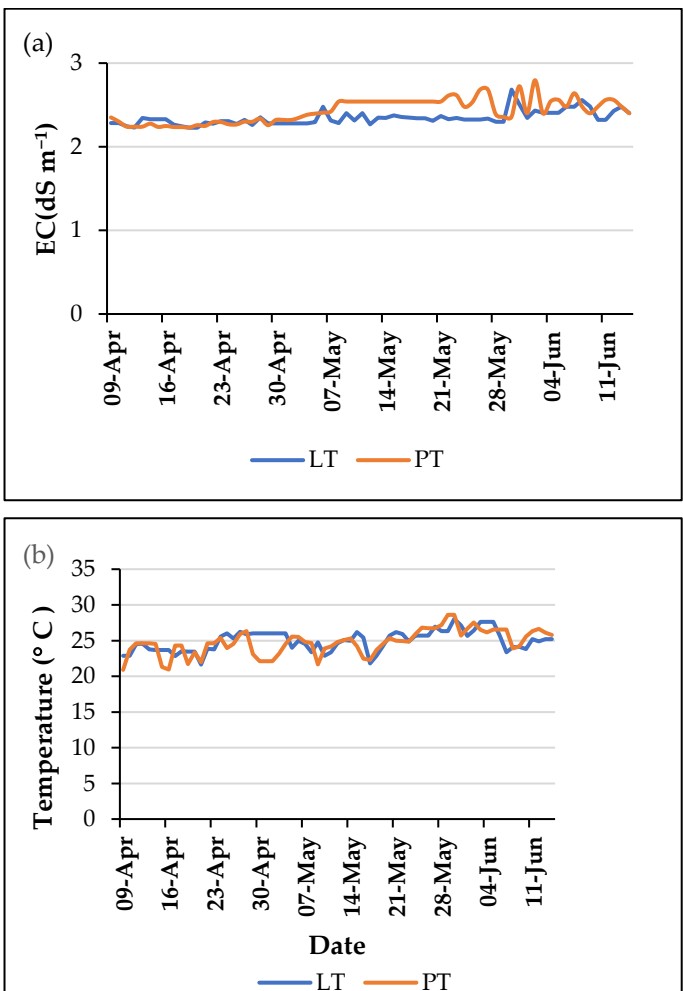

**Figure 4.** Nutrient solution electrical conductivity (**a**) and temperature (**b**).

*2.6. Measurements*

2.6.1. Plant Growth, Development, and Dry Matter Partitioning

Plant growth parameters such as plant height and the number of nodes per plant were measured at the end of the growing period.

Additionally, the plant leaf area index (LAI) was calculated weekly from 60 days after transplantation (DAT) to the end of the growing period using the method described by Ahn et al. [18]. In each treatment, five selected plants were subjected to the LAI evaluation. First, 30 leaves having different sizes were pruned. The images of the leaves were taken with a digital camera, and their areas were estimated using image j software. The length (L) and width (W) of the leaves were also measured with a ruler. An equation was developed to estimate the leaf area from L and W. Next, ten randomly selected leaves' lengths and widths were measured on each plant with a ruler. The leaf area index was calculated as follows:

$$Leaf\ area\ index = \frac{Average\ single\ leaf\ area\ \times\ Average\ number\ of\ leaf\ per\ plant\ \times\ Number\ of\ plant}{Planted\ area}$$

The fresh weights of all leaves and stems pruned during the experiment were measured. At the end of the experiment, five (5) selected plants of each treatment were destructively harvested and separated into leaves, stems, and roots, and their fresh weights were measured. In addition, all leaves, stems, roots, and sampled fruits collected during and at the end of the experiment were oven-dried at 80 °C for seven days, and their dry weights were measured.

### 2.6.2. Water and Nutrient Use Efficiencies

Water consumption of the cucumber plants was measured with a water counter. Water use efficiency (WUE) was determined as a ratio of fruit yield per plant to total water uptake per plant [19].

Samples of nutrient solutions were collected weekly and analysis of concentrations of mineral elements was carried out. The samples were analyzed using Dionex ICS1100 ion chromatography (Thermo Fisher Scientific Inc., Waltham, MA, USA).

The nutrient use efficiency (NUE) was determined as a ratio of the fruit yield per plant to the total nitrogen ($NO_3$-N and $NH_4$-N) uptake per plant [20].

### 2.6.3. Plant Physiology

Individual leaf photosynthetic rate was measured at 60 DAT using the LI-6400 (LI-COR, Lincoln, NE, USA). Measurement was carried out on mature leaves located at the bottom, middle, and upper part of five selected plants' canopies in each treatment between 10:00 AM and 2:00 PM. The temperature, relative humidity, and $CO_2$ concentration in the leaf clear top chamber were kept at the ambient environmental conditions of the greenhouse [21].

### 2.6.4. Plant Productivity and Fruit Quality

Each fruit was tagged with the anthesis date during the experiment. Fruits were harvested once their weight reached approximately 100 g. At each harvest, the days to harvest, fruit weight (g), fruit length (cm), fruit diameter (cm), fruit shape, and fruit appearance were recorded. Moreover, the number of fruits per plant, fruit yield per plant, per meter square, and marketable yield were calculated.

### 2.6.5. Workload

The times spent on main activities such as old leaves removal, training, and harvesting were recorded in the two treatments.

### 2.6.6. Data Analysis

The collected data were analyzed by ANOVA and comparison test between means using XL STAT software Ver. 2022.1.2 (XLSTAT statistical and data analysis solution. New York, NY, USA).

## 3. Results

### 3.1. Characteristics of the Greenhouse's Environmental Parameters

The daily cumulative indoor irradiation progressively increased from February to June with an average value of 8.6 MJ m$^{-2}$ (Figure 5a). The daily average temperature, relative humidity, and $CO_2$ concentration (Figure 5b) were 20.8 °C, 70.4%, and 497.2 µmol mol$^{-1}$, respectively.

The statistical analysis results indicated that the training methods affected the air temperature near the plant's canopy. The average daytime air temperatures were 25.8 and 26.1 °C in the LT and PT treatments, respectively. On the other hand, nighttime air temperatures were 17.8 °C in the LT and 18.3 °C in the PT.

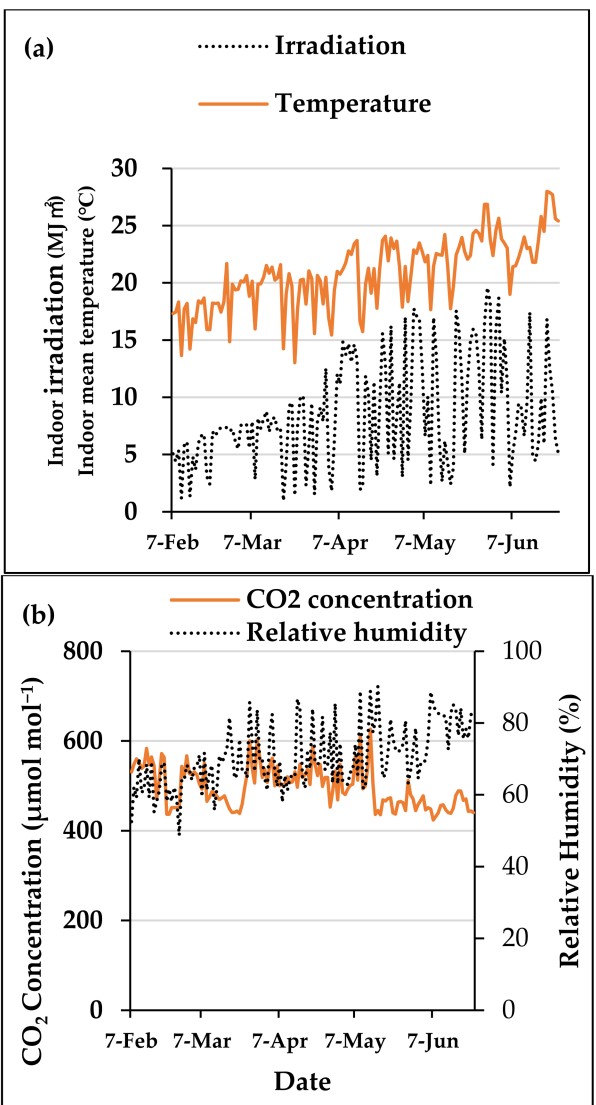

**Figure 5.** Daily mean temperature and irradiation (**a**), and daily average relative humidity and $CO_2$ concentration (**b**) in the greenhouse.

### 3.2. Plant Growth, Development, and Dry Matter Partitioning

The total stem length of each plant and the number of nodes per plant, determined at the end of the experiment, were significantly different in the two training methods. The highest total stem length (10.9 m) and the highest number of nodes per plant (133) were found in the LT treatment (Table 2).

**Table 2.** Effect of training methods on cucumber plant growth.

| Treatment | Main Stem Length (m) | Side Shoots Length (m) | Stem Total Length (m Plant$^{-1}$) | Number of Nodes per Plant |
|---|---|---|---|---|
| LT | 1.5 [a] | 9.4 [a] | 10.9 [a] | 133 [a] |
| PT | 1.6 [a] | 4.9 [b] | 6.5 [b] | 86 [b] |
| Significance | NS | ** | ** | ** |

Treatment effects were significant at 1% (**) probability level or were not significant (NS). Different letters indicate significant differences at $p < 0.05$, according to Tukey's multiple comparison test.

The change in the leaf area index (LAI) in the two treatments is shown in Figure 6.

**Figure 6.** Leaf area index in the two treatments from 7 April 7 to 26 May. Within each date, the same letters are not significantly different at $p < 0.05$ by Tukey's multiple comparison test.

From 21 April (73 DAT) to 19 May (101 DAT), the LAI was significantly higher in the LT treatment. On the other hand, less variability of the LAI was found in the PT treatment (2.7–3.3 $m^2$ $m^{-2}$) compared to the LT treatment (2.9–4.8 $m^2$ $m^{-2}$). The number of leaves per plant was identified as the major contributing factor to the difference in the LAI between the two treatments (data not shown).

The proportional dry matter distribution between fruits and vegetative parts is presented in Table 3. The total dry matter produced per plant (TPDM) was significantly higher in the lowering training treatment. The plants cultivated under the LT had more dry mass of leaves, stems, fruits, and roots than those grown with the PT method. However, 41.02% of the total dry matter produced per plant was partitioned to fruits in the PT treatment, while 36.2% of the TPDM was allocated to the fruits in the LT treatment. Additionally, the fruit dry matter production was significantly higher in the early harvest stage in the PT treatment.

**Table 3.** Effects of training methods on leaf, stem, fruit, and roots dry weight and dry matter distribution to plant organs.

| Variable | Treatment | Days after Transplantation | | | | | Dry Matter Partitioning (%) |
|---|---|---|---|---|---|---|---|
| | | 28 | 56 | 84 | 112 | 133 | |
| Leaf Dry Matter (g plant$^{-1}$) | LT | 0.5 [b] | 19.0 [b] | 34.2 [b] | 115.2 [a] | 448.1 [a] | 51.3 [a] |
| | PT | 1.1 [a] | 27.1 [a] | 53.1 [a] | 77.5 [b] | 339.3 [b] | 47.8 [b] |
| | Significance | *** | *** | *** | *** | *** | *** |
| Stem Dry Matter (g plant$^{-1}$) | LT | 0.051 [b] | 2.1 [b] | 2.4 [b] | 2.4 [b] | 87.7 [a] | 10.1 [a] |
| | PT | 0.175 [a] | 4.1 [a] | 4.9 [a] | 4.9 [a] | 62.9 [b] | 8.9 [b] |
| | Significance | *** | *** | *** | *** | *** | *** |
| Fruit Dry Matter (g plant$^{-1}$) | LT | | 22.9 [a] | 136.1 [a] | 264.6 [b] | 318.8 [a] | 36.20 [b] |
| | PT | | 36.8 [b] | 138.8 [a] | 238.2 [a] | 294.3 [a] | 41.02 [a] |
| | Significance | | *** | NS | * | NS | *** |
| Root Dry Matter (g plant$^{-1}$) | LT | | | | | 21.0 [a] | 2.40 [a] |
| | PT | | | | | 16.6 [b] | 2.28 [a] |
| | Significance | | | | | ** | NS |
| Plant Dry Matter (g plant$^{-1}$) | LT | | | | | 875.6 [b] | |
| | PT | | | | | 713.1 [a] | |
| | Significance | | | | | *** | |

Treatment effects were significant at 5% (*) or 1% (**) or 0.1% (***) probability level or were not significant (NS). Different letters indicate significant differences at $p < 0.05$, according to Tukey's multiple comparison test.

### 3.3. Water and Nutrient Use Efficiencies

The water uptake of the cucumber plant was influenced by the greenhouse environmental factors (Table 4). The temperature and solar radiation significantly correlated with the plants' water uptake in the LT treatment. The correlation between solar radiation and water uptake was significant in the PT treatment. Solar radiation showed the highest r value in both treatments.

**Table 4.** Correlation between greenhouse environmental factors and water uptake.

| Parameter | *n* | LT | | | | PT | | | |
|---|---|---|---|---|---|---|---|---|---|
| | | Regression Equation | R | *p*-Value | Significance *p* < 0.05 | Regression Equation | R | *p*-Value | Significance *p* < 0.05 |
| | | *x*-axis water uptake | | | | | | | |
| | | *y*-axis | | | | | | | |
| solar radiation | 19 | y = 2.6792x + 5.6901 | 0.68 | 0.001 | *** | y = 2.8672x + 5.9714 | 0.63 | 0.004 | ** |
| Temperature | 19 | y = 1.6331 x+18.903 | 0.5 | 0.03 | * | y = 1.4798x + 19.318 | 0.4 | 0.09 | ns |
| Relative humidity | 19 | y = 2.6069x +67.351 | 0.3 | 0.22 | ns | y = 1.9865x + 68.357 | 0.19 | 0.43 | ns |

Correlations were significant at 5% (*) or 1% (**) or 0.1% (***) probability level or were not significant (NS).

The weekly water consumption in the two treatments is shown in Figure 7a. The water uptake varied during the 19 weeks of the cropping period. Crop water use was considerably lower during the early vegetative stage, increased progressively, and reached a peak during the 9th week (63–70 DAT) and 11th week (77–84 DAT) after transplantation in the PT and LT treatments, respectively. The peak periods coincided with the flowering and fruiting stages of the crop in the two treatments.

From the peak period to the end of the experiment, the water uptake remained higher in the LT treatment and fluctuated in both treatments. Four (4) weeks before the end of the experiment, the water uptake declined continuously in all treatments.

A total of 143.6 and 121.2 L of water were used to meet each cucumber plant's water requirement in the LT and PT treatments, respectively, during the cropping period. Statistical analysis results indicated that WUE was not significantly different between the treatments. However, NUE was considerably higher (32.6%) in the PT treatment (Table 5).

**Table 5.** Effect of training methods on water and nutrient use efficiency at the end of the experiment (133 DAT).

| Treatment | Water Uptake per Plant (L Plant$^{-1}$) | Nitrogen Uptake per Plant (kg Plant$^{-1}$) | WUE (kg kg$^{-1}$) | NUE (kg kg$^{-1}$) |
|---|---|---|---|---|
| LT | 143.6 | 0.028 | 0.065 [a] | 308.7 [b] |
| PT | 121.2 | 0.018 | 0.068 [a] | 409.2 [a] |
| significance | | | NS | *** |

Treatment effects were significant at 0.1% (***) probability level or were not significant (NS). Different letters indicate significant differences at *p* < 0.05, according to Tukey's multiple comparison test. NUE = Fruits yield(kg)/Nitrogen uptake(kg); WUE = Fruits yield(kg)/Water uptake (kg)

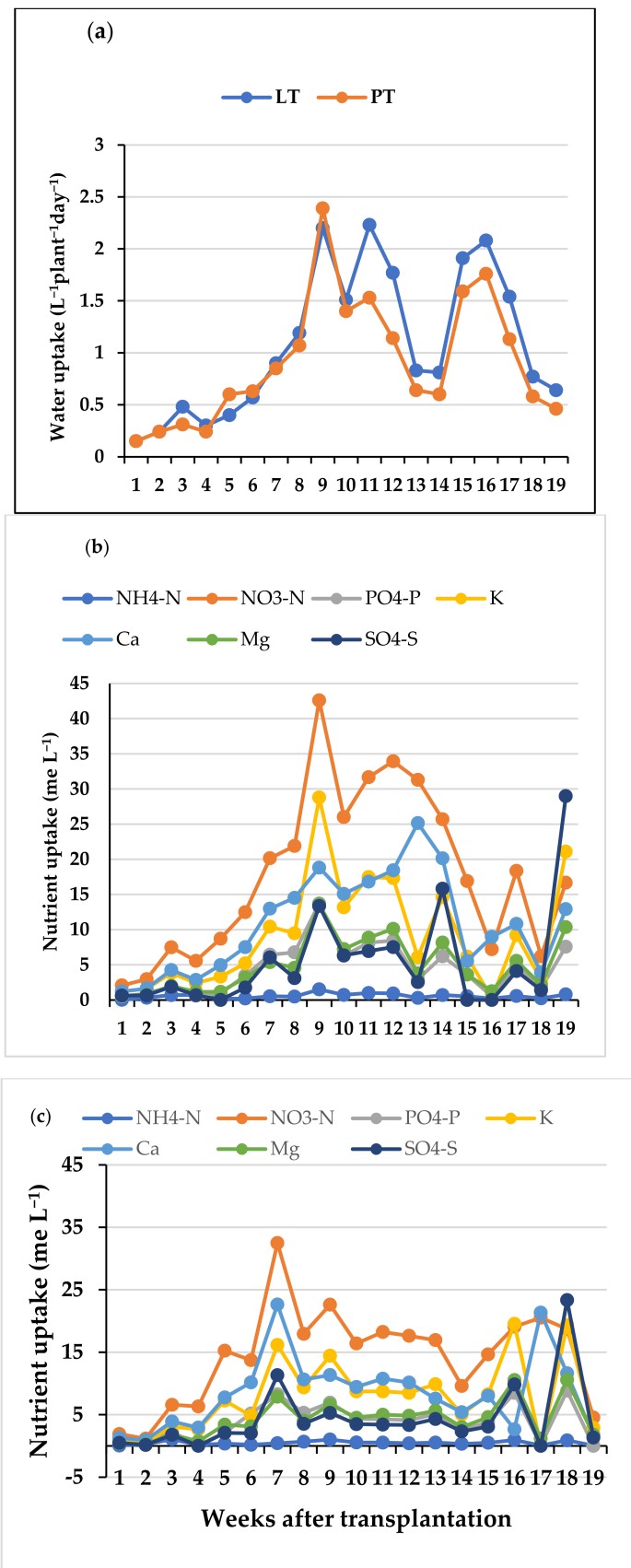

**Figure 7.** (**a**) Water uptake in the two treatments, (**b**) nutrient uptake in the LT treatment, (**c**) nutrient uptake in the PT treatment.

Figure 7b,c shows the nutrient uptake characteristics during the 19 weeks of the cropping period in the two treatments. The training methods exerted an effect on the nutrient absorption capacity of the cucumber plants. The cumulative nutrient uptake of all nutrients was higher in the LT treatment. Moreover, the peak period of $NO_3$-N and K uptakes in the LT treatment was nine weeks after transplantation.

In the pinching treatment, the uptake peak of $NO_3$-N and Ca was recorded seven weeks after transplantation. The lowest absorbed element in both treatments was $NH_4$-H.

### 3.4. Plant Physiology

The individual leaf photosynthetic rate between the two treatments was not significantly different (Figure 8). However, in the LT treatment, the photosynthetic rate of leaves gradually decreased (from top to bottom) following the leaves' positions on the lateral branches. In contrast, in the PT treatment, it tended to be higher in the middle part of the plant canopy.

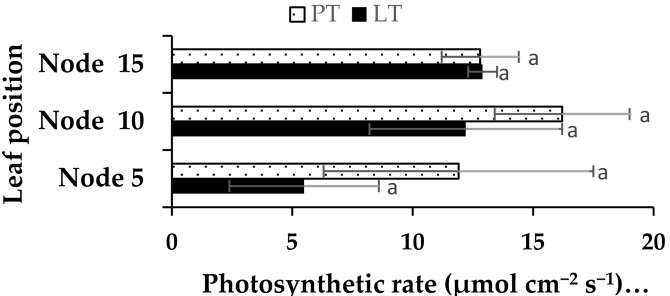

**Figure 8.** Effect of training methods on photosynthetic rate of individual leaf. Within each node, the same letters are not significantly different at $p < 0.05$ by the Tukey's multiple comparison test.

### 3.5. Plant Productivity and Fruit Quality

The harvest started on 21 March and ended on 23 June. The yield variation in the two treatments during the harvesting period is shown in Figure 9a. The plants of the PT produced a relatively higher yield per plant at the early harvesting stage. However, at the mid and late harvesting stages, the fruit yield per plant was higher in the LT treatment.

As shown in Figure 9b, the total fresh weight of fruits produced per plant was significantly higher in the pinching treatment from the 6–10th week after transplantion. However, from the 11–12th week after transplantation, the fruit yield per plant was similar in the two treatments.

After the 12th week after transplantation, the total fresh weight of fruits produced per plant became higher in the LT treatment.

The highest total yield (15.4 kg m$^{-2}$) and marketable yield (13.8 kg m$^{-2}$) per unit area were recorded in the LT treatment.

Similarly, a significantly higher total number of fruits and the number of marketable fruits per plant were also obtained in the same treatment (Table 6). Additionally, the abnormal fruit rate per plant was not significantly different among the two treatments.

**Table 6.** Effect of training methods on cucumber plant yield and yield components.

| Treatment | Total Yield | | Marketable Yield | | Number of Fruits/Plants | | Non-Marketable Fruits per Plant (%) |
|---|---|---|---|---|---|---|---|
| | kg Plant$^{-1}$ | kg m$^{-2}$ | kg Plant$^{-1}$ | kg m$^{-2}$ | Total | Marketable | |
| LT | 9.3 [a] | 15.4 [a] | 8.3 [a] | 13.8 [a] | 84.9 [a] | 75.7 [a] | 10.8 [a] |
| PT | 8.2 [b] | 13.6 [b] | 7.4 [b] | 12.4 [b] | 71.3 [b] | 64.9 [b] | 8.9 [a] |
| Significance | * | * | * | * | *** | ** | NS |

Treatment effects were significant at 5% (*) or 1% (**) or 0.1% (***) probability level or were not significant (NS). Different letters indicate significant differences at $p < 0.05$, according to Tukey's multiple comparison test.

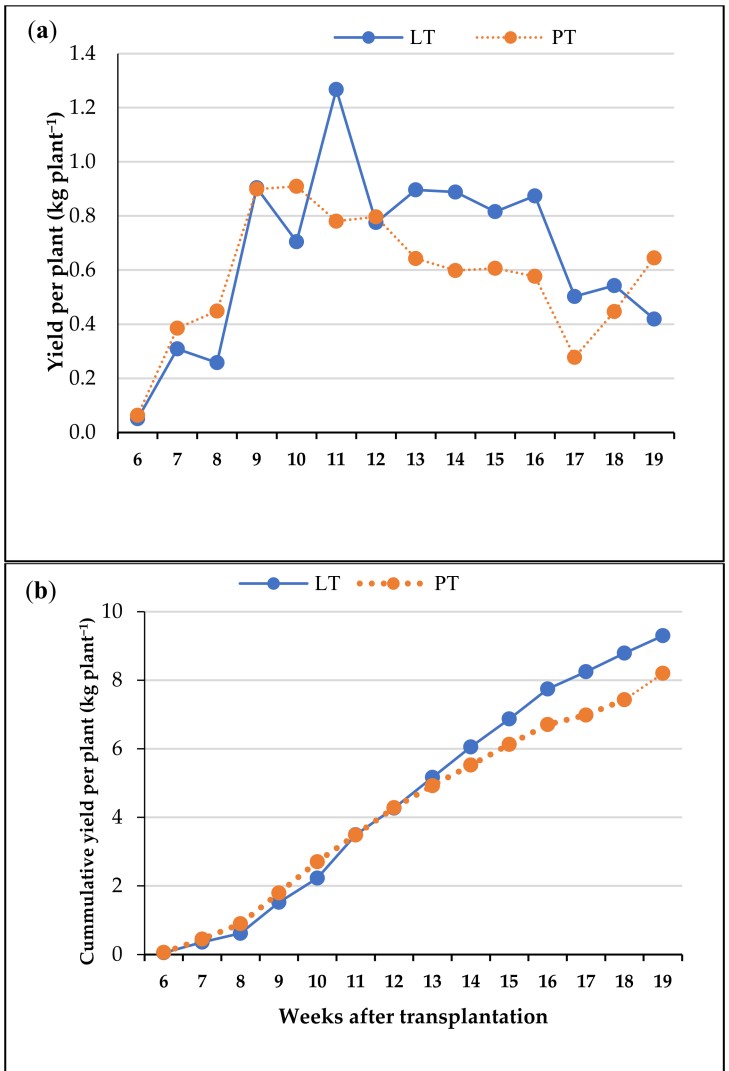

**Figure 9.** Plant yield variation (**a**) and plant cumulative production of fruits (**b**) during the harvesting period.

Harvested fruits weighted, on average, 114.7 g and 109.1 g in the PT and LT treatments, respectively (Table 7). Those fruits were significantly longer in the PT treatment. On the other hand, fruits' diameters were found significantly higher in the LT treatment. The length a/b ratio, which is the ratio of the sizes of the two sides of the fruits, indicated that fruits were more straight in the PT treatment.

**Table 7.** Effect of training methods on fruit shape, size, and dwell time.

| Treatment | Length a (cm) | Length b (cm) | Length b/a | Diameter (mm) | Fruit Weight (g fruit⁻¹) | Days to Harvest |
|---|---|---|---|---|---|---|
| LT | 21.6 [b] | 19.0 [b] | 0.88 [b] | 28.6 [a] | 109.1 [b] | 18.9 [a] |
| PT | 23.1 [a] | 20.6 [a] | 0.90 [a] | 28.3 [b] | 114.7 [a] | 14.1 [b] |
| Significance | *** | *** | *** | ** | *** | *** |

Treatment effects were significant at 1% (**) or 0.1% (***) probability level. Different letters indicate significant differences at $p < 0.05$, according to Tukey's multiple comparison test.

The dwell time of fruits on the plant was estimated to be shorter in the PT treatment, as fruits spent, on average, 14.1 days to get standard harvest weight in that treatment.

In the LT treatment, the average number of days from flowering to harvest was 18.9.

### 3.6. Workload

The workloads for harvest, old leave removal, and lateral shoot training were recorded from the pinching of the main stem (30 DAT) to the end of the experiment. The statistical analysis results indicated a significant difference between the treatments for all variables (Table 8). On average, 22.4 and 25.8 s were spent to harvest a single fruit in the LT and the PT treatments, respectively. Similarly, the time spent for a single leaf removal was 11.4 s in the LT treatment and 20.3 s in the PT treatment. The times for a single fruit harvest and a single leaf removal were 13.2% and 43.8%, respectively, lower in the LT treatment. In contrast, the time spent to train the lateral vines of a single plant was 89.5% higher in the LT treatment.

**Table 8.** Workload and workload partitioning.

| Treatment | Leaf Removal (Second Leaf$^{-1}$) | Fruit Harvest (Second Fruit$^{-1}$) | Side Shoots Training (Minutes Plant$^{-1}$) | Total Workload (Minute Plant$^{-1}$) | Workload Partitioning per Plant (%) | | |
|---|---|---|---|---|---|---|---|
| | | | | | Leave Removal | Harvest | Training |
| LT | 11.4 [b] | 22.4 [b] | 10.8 [a] | 56.9 [a] | 25.2 [b] | 55.6 [b] | 19.2 [a] |
| PT | 20.3 [a] | 25.8 [a] | 5.7 [b] | 51.9 [b] | 30.0 [a] | 59.0 [a] | 11.0 [b] |
| Significance | * | * | *** | * | *** | * | *** |

Treatment effects were significant at 5% (*) or 0.1% (***) probability level. Different letters indicate significant differences at *p* < 0.05, according to Tukey's multiple comparison test.

The total time spent on a plant for fruit harvest, old leaves removal, and lateral vines training was 9.6% higher in the LT treatment compared to the PT treatment.

## 4. Discussion

### 4.1. Plant Growth, Development, and Dry Matter Partitioning

In this study, plant total stem length and the number of nodes per plant were significantly higher in the LT treatment. In addition, the leaf area index was higher in the same treatment in the middle stage of the cropping period.

The reduction of plant height and leaf area index in the PT treatment could be attributed to the successive pinching shocks received by the lateral shoots, which could suppress roots and shoot growth. In contrast, lateral shoots continued to grow in the LT treatment and did not receive any pinching shock.

The effect of training methods on plant growth characteristics observed in the current study is consistent with many previous findings. For example, Higashide et al. [22] reported that the cucumber grown under LT conditions developed a higher stem length and number of leaves than the cucumber cultivated using the PT method. In addition, Premalatha et al. [14] reported a significant effect of cucumber training methods on the leaf area index.

Krishnaveni et al. [23] observed a considerable decrease in plant height with an increased number of pinching on fenugreek (*Trigonella foenum-graecum* L.) and indicated that the plant height reduction in the pinching treatment could be due to suppressed root and shoot growth. The suppression of root and shoot growth in the pinching treatment was confirmed in the current study due to the lower dry matter production of those plant's organs recorded in that treatment (Table 3). The total dry matter production per plant was higher in the LT treatment. This result could be explained by the difference in light use efficiency and interception between the two training methods. Iwasaki et al. [24] compared the two training methods on several cucumber cultivars and concluded that the total

dry matter production correlated with the light interception at 40 DAT and the light use efficiency during the entire experimental period.

### 4.2. Water and Nutrient Use Efficiencies

The water consumption in the early growth stage in this study is consistent with the findings of Zotarelli et al. [25], who reported that at the early growth stages of cucumber, water uptake capacity by roots is limited.

The correlation between the water uptake and the environmental factors indicated that solar radiation is the main factor influencing plant water consumption. This result is in line with the findings of Salas et al. [26], who reported that among environmental factors, solar radiation could be considered the main factor of water absorption during the day. In addition, Gislerod and Adams [27] studied the uptake of water and potassium by cucumber and tomato. They reported that the uptake of both water and potassium increased in response to solar radiation in the two crops.

The peaks of the water uptake in both treatments were recorded during the peak of the harvested fruits. Moreover, the LAI and the solar radiation were high during the two treatment peaks of water uptake.

Schwarz and Kuchenbuch [28] indicated a relation between daily total solar radiation and the water uptake rate after the beginning of tomato fruit harvest. The same authors said that the water uptake rate depends on the plant growth stage, which was observed in the current study.

In this experiment, the higher water uptake recorded in the LT treatment, from week 10th to the end of the experiment, is probably caused by the high plant total biomass (stem, leaves, and root) produced in that treatment. The higher roots and leaves fresh weight produced in the LT treatment promoted water and nutrient uptake. On the other hand, the nutrient use efficiency was significantly lower in the LT treatment. This finding can be explained by producing vegetative parts instead of fruits in that treatment (Table 3).

### 4.3. Plant Productivity and Fruit Quality

The significantly higher total yield observed in the LT treatment could be explained by the high number of fruits per plant recorded in that treatment. Moreover, in the LT treatment, the plant had a significantly higher number of nodes to initiate flowers and a higher leaf area index to intercept more sunlight. These two factors contributed to enhance plants' productivity. The higher total yield recorded in the LT treatment in this study is consistent with the yield observed by Hirama et al. [29]. However, this result is inconsistent with the results of Isomura et al., Sakata et al., and Higashide et al. [22,30,31], who reported that the PT method had a significantly higher yield than the LT method.

Sakata et al. [31] analyzed the two training methods and indicated that the lower yield observed in the LT is caused by the loss of yield from the first lateral branches. They added that the lowering training method might not produce the best yield in their experiment and suggested future investigation on that training method.

The low yield observed in the previous studies mentioned above could be attributed to the shorter cultivation period of their experiments. However, as observed in the current study, the early yield was significantly higher in the PT treatment. Therefore, if the cultivation period is shorter, the PT method will be more productive because the difference in the early yield cannot be recovered since it takes time to trim the lateral branches in the LT method.

Apart from the cultivation duration, to obtain a higher yield using the LT method, predominantly female-type cultivars are required because the LT method involves setting the fruits on the main stem for an extended period [11]. In addition, the wire height is essential to increase the yield in the LT method. In our study, the wire height was set at 2.5 m above the ground, higher than the wire height of the previous investigations illustrated above. Increasing the wire height exposes the plant's canopy to more sunlight.

Concerning the fruit quality, the percentage of abnormal fruits per plant was not significantly different between the training methods. However, the marketable yield was significantly higher in the LT treatment due to the high number of fruits harvested. A similar result was observed by Yatomi and Ono [32], who reported an increase in the percentage of marketable fruits by using the lowering training method with four lateral branches per plant. Moreover, the dwell time of fruits on the plant and the fruits' shapes were significantly different between the two training methods. These results could be explained by the difference in environmental conditions, such as air temperature, air velocity, and irradiation around the fruits in the two treatments. In addition, the environmental conditions around the fruits were probably affected by the difference in the leaf area index in the two treatments.

As reported by Iwasaki et al. [33], the light distribution was probably more uniform across the plant canopy in the pinching treatment. In contrast, the lower part of the canopy received less sunlight in the LT treatment. Data recorded inside the plant canopy showed that the average day and night temperatures were significantly different between the two treatments (data not shown). The highest average day and night temperatures (26.1 °C and 18.3 °C, respectively) were recorded in the pinching treatment, accelerating the fruit growth rate. Even though the differences between the treatments in the day and night temperatures are minimal (0.3 and 0.5 °C respectively) due to the small plant population of the experiment, it is essential to note that these differences and the differences in the other environmental factors could increase in the case of large-scale cultivation.

The fruit size (length and diameter) was also significantly different in the two treatments. As fruit length appeared to be a varietal character of some fruit-vegetables such as bell pepper and cucumber [34,35], the difference in fruit length observed in this study could be attributed to the difference of individual fruit weight at harvest in the two treatments. Previous studies reported that the training methods do not affect fruit length [9,25]. However, cucumber fruit diameter is a character that can be affected by training methods. Premalatha et al. [14] observed the influence of training methods on cucumber fruit diameter and reported that high LAI could favorably affect the character.

Figure 10 shows the pictures of the cultivated cucumbers.

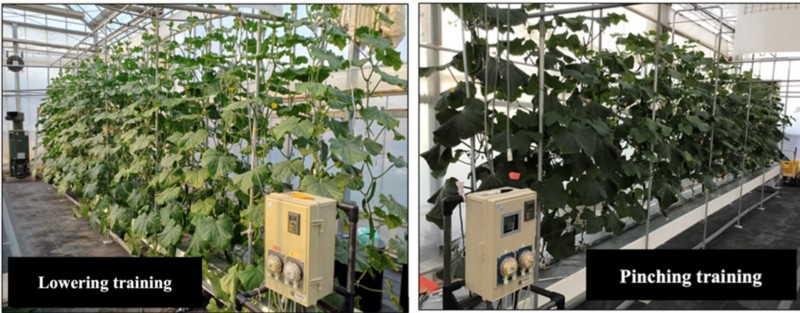

**Figure 10.** Pictures of cucumber grown using pinching and lowering training methods.

*4.4. Strengths and Weaknesses of the Training Methods*

Table 9 summarizes the strengths and weaknesses of each training method and proposes solutions to improve the weaknesses.

**Table 9.** Strengths and weaknesses of training methods and how to improve weaknesses.

| Training Method | Strengths | Weakness | Solutions to Improve |
|---|---|---|---|
| Lowering | Plant management is easy to understand by farmers. * Employment type (part-time workers) can be introduced | Low initial yield | Manage the plant shape to promote early fruit setting. |
| | | | Extend the cultivation period (5–6 months) |
| | | | Raise the wire height to increase the amount of intercepted irradiation |
| | Maintaining three lateral branches * High yield for long-term cultivation (5–6 months) | Basal leaves touch the ground and can cause diseases | Use tools (clips and ropes) to hold the side shoots at a certain height from the ground |
| | | High percentage of abnormal fruits | Control plant vigor through the nutrient solution supply management |
| Pinching | High yield for short-term cultivation (3–4 months) | Plant management is difficult to understand by farmers. * Yield varies depending on farmers' experience | Popularize the cultivation method through the means of audio-visual communication tools |

*4.5. Prospects*

As described above, each training method has its strengths and weaknesses. For example, the lowering training method is suitable for long-term cultivation, while the pinching training method is profitable for short-term cultivation.

The lowering training method is indicated to maintain enterprise productivity for large and small enterprises with a labor shortage as a limiting factor.

Furthermore, apart from being easy to perform, the lowering training method offers the possibility to ease some works, such as harvesting fruits and removing old leaves. This statement has been proved in the current experiment through the workload recorded for those works and is consistent with the results of Yatomi and Ono [32].

Since the position of matured fruits and old leaves is fixed in the lowering training method, the harvest and the leaves removal become easy. Furthermore, the fact that humans can harvest fruits and remove old leaves more efficiently also means that the lowering training method has a high affinity for removing old leaves and harvesting fruits by robots. Therefore, from a long-term perspective, it seems reasonable to spread the lowering training method to ease the implementation of cucumber fruit harvest by robots currently being tested in smart agriculture in Japan [10].

**5. Conclusions**

In this study, cucumber plants cultivated using the lowering training method produced a higher total and marketable yield than plants grown with the pinching training method. This result was attributed to the higher number of nodes per plant and increased intercepted light due to the higher LAI recorded in the lowering training treatment. Furthermore, workloads of old leaf removal and fruit picking were reduced in the lowering training treatment due to the fixed position of old leaves and mature fruits in that treatment. On the other hand, the nutrient use efficiency was lower in the LT treatment due to the production of more leaves instead of fruits. Therefore, to improve the yield and nutrient use efficiency, we recommend reducing the leaf area index in that treatment through quantitative nutrient supply management.

**Author Contributions:** Conceptualization, N.S., S.T. and A.N.; data curation, N.S.; formal analysis, N.S., S.T. and A.N.; funding acquisition, S.T.; investigation, N.S., S.T. and A.N.; methodology, N.S., S.T., A.N. and O.N.; project administration, S.T. and A.N.; resources, S.T. and O.N.; software, N.S.; supervision, S.T. and A.N.; validation, S.T. and A.N.; writing—original draft, N.S.; writing—review and editing, N.S., S.T. and A.N. All authors have read and agreed to the published version of the manuscript.

**Funding:** This research received no external funding.

**Data Availability Statement:** Data are contained within the article.

**Acknowledgments:** Authors are grateful to Japan International Cooperation Agency (JICA) for all kinds of support received through its Agriculture Studies Networks for Food Security (Agri-Net) program.

**Conflicts of Interest:** The authors declare no conflict of interest.

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
