# Peer review of "Effective Training Methods for Cucumber Production in Newly Developed Nutrient Film Technique Hydroponic System"

_horticulturae, doi:10.3390/horticulturae9040478_

Round 1

Reviewer 1 Report

The manuscript is written well overall. I think it can be accepted with this form.

Author Response

Dear Reviewer, 

Thank you for reading our manuscript and for your comments. 

Best regards, 

Reviewer 2 Report

GENERAL COMMENTS:

The paper aims to evaluate the most suitable cucumber crop formation method for production in a NFT hydroponic system. It compares agronomic results using two plant formation schemes by pruning and training. This is a subject that has been studied previously, as shown in the bibliography, but necessary to optimise the crop productivity and it is well justified. The results obtained are of interest for transfer to farmers.  

The manuscript is clear and its structure adequate. Some specific comments are given below.

SPECIFIC COMMENTS:

-      Line 90. Include the greenhouse dimensions.

-      Lines 135-136. Only two culture lines have been used in the study, each dedicated to one treatment. What is their orientation?. What is the separation between the two?. Their arrangement in the greenhouse could lead to an effect of the position on the parameters to be evaluated?.

-      Lines 192-193. Since the reference describing the LAI estimation method is in Japanese, it would be convenient to describe the methodology in the paper.

-      Line 208-212. This section is not sufficiently well described. It is essential to know the incident radiation on the leaf during the measurement. How many repetitions are taken for each location, on different leaves?. On the other hand, these determinations should provide interesting data to be included in the results section; however, it has been omitted to show the data that support the statement made in lines 304-308. Provide such data.

-      Line 217-218. Regarding the evaluation of fruit production, each plant in the system (except the borders) was considered a replicate? So there were 15 repetitions; if not, please explain.

-      Lines 245-247. The LAI evolution shows differences between treatments in values that could have reached the critical LAI, thus not improving radiation interception. A study on training should include measurements of the interception of radiation by the crop, since this is a determining factor of production. Are such data available to provide?

-      Table 3. The table includes biomass data on different days throughout the cultivation, however in Material and methods it is indicated that a determination was made at the end of the experiment. Where are these results obtained from? These determinations are destructive and if the number of plants per treatment at the beginning of the crop was 15 (excluding the edges), how can it be explained that there were 5 sampling dates?

-      Lines 268-274. The correlation between water absorption and radiation is not new, it is well documented in the scientific literature and its relationship with vapour pressure deficit (VPD) and LAI is also reported (Medrano et al. 2005. Scientia Horticulturae 105: 163–175).  These correlations could be incorporated in table 4. Was the correlation different for each treatment?

-      Figure 7- Correct the name of the treatment HWT to LT.

-      Lines 300-301. Use the same y-axis scale in figures 7 b and c.

-      Figure 8. Change the x-axis to weeks after transplantation or date to unify with the rest of the figures.

-      Lines 323-324 and Table 6. If the density is 0.6 plants m-2 (Line 144) then the production values per square meter should be lower than those obtained per plant. Check them.

-      Lines 354-373: There is no discussion of the dry matter results.

-      Line 429-433 and Line 436-440: Provide in the Results section the microclimate values recorded around the canopy of both treatments to justify the statement.

-      Could the microclimatic differences in the canopy environment in each treatment be related to the position of the treatment in the greenhouse? As there is only one crop line per treatment, there could be a clear influence of the position regarding the incidence of radiation. It is important to consider it in the discussion.

Author Response

Dear Reviewer,

Thank you for the time spent on my manuscript. Your comments and remarks were relevant. 

Please find attached the responses to your questions and comments.

Best regards,  

Reviewer 3 Report

The ms contains serious mistakes.

The introduction lacks references; materials and methods need to be improved; the figures are not clear.

The style of the reference section is wrong.

Author Response

Dear Reviewer,

Thank you for the time you spent on our manuscript. Your comments and remarks were relevant. 

I attached the responses to your questions and comments.

Best regards,  

Round 2

Reviewer 2 Report

In the revised version of the manuscript, the statistical results of the photosynthesis data presented show no significant differences between treatments even though the data are quite different. This indicates that there may be excessive variability between replicates. The standard deviation of the data should be provided. 

Author Response

Dear Reviewer, 

Thank you for your feedback. Your remarks contributed to improving the quality of the manuscript. In response to your comment, I provided the standard deviation of the data in figure 8. 

Best regards, 

Reviewer 3 Report

Endorsed

Author Response

Dear Reviewer,

Thank you for your constructive comments during the peer review process. It contributed to improving the quality of the manuscript. 

Best regards.